# Particle Swarm Algorithm Path-Planning Method for Mobile Robots Based on Artificial Potential Fields

**DOI:** 10.3390/s23136082

**Published:** 2023-07-01

**Authors:** Li Zheng, Wenjie Yu, Guangxu Li, Guangxu Qin, Yunchuan Luo

**Affiliations:** 1School of Automation and Electrical Engineering, Chengdu Technological University, Chengdu 611730, China; 2School of Automation, Chengdu University of Information Technology, Chengdu 610225, China; 3Chengdu Shengke Information Technology Co., Ltd., Chengdu 610017, China; 4Sichuan Research Institute of Chemical Quality and Safety Inspection, Chengdu 610031, China; scyqhgcyjl@163.com

**Keywords:** mobile robot, route planning, PSO, artificial potential field

## Abstract

Path planning is an important part of the navigation control system of mobile robots since it plays a decisive role in whether mobile robots can realize autonomy and intelligence. The particle swarm algorithm can effectively solve the path-planning problem of a mobile robot, but the traditional particle swarm algorithm has the problems of a too-long path, poor global search ability, and local development ability. Moreover, the existence of obstacles makes the actual environment more complex, thus putting forward more stringent requirements on the environmental adaptation ability, path-planning accuracy, and path-planning efficiency of mobile robots. In this study, an artificial potential field-based particle swarm algorithm (apfrPSO) was proposed. First, the method generates robot planning paths by adjusting the inertia weight parameter and ranking the position vector of particles (rPSO), and second, the artificial potential field method is introduced. Through comparative numerical experiments with other state-of-the-art algorithms, the results show that the algorithm proposed was very competitive.

## 1. Introduction

With the rapid development of science and technology, mobile robotics has been widely applied in many fields due to its high efficiency, independence from environmental restrictions, and strong anti-interference ability [1]. In practical environments, when a mobile robot receives a given task, it must be able to move autonomously from the target starting point to the target endpoint and avoid obstacles to find the shortest path. This makes the path planning of mobile robots one of the key research problems in robot applications [2].

The feasibility of mobile robot path planning is determined by three factors: the accuracy of the environment, the accuracy of the robot’s positioning, and the certainty of the number and locations of the obstacles [3]. When a mobile robot decides its movement, it needs to plan the optimal path according to its task, saving as much time, energy, distance, etc., as possible. The most necessary capability of a mobile robot in path planning is the ability to enable the robot to plan a path in its surrounding environment that prevents collision with other objects and accomplishes the task of moving from the target starting point to the target ending point. If a reasonable path can be planned, it can greatly improve its efficiency and reduce energy consumption [4].

Robot path planning can be divided into single-robot and multi-robot path planning according to the number of robots. The path-planning problem for mobile robots can be formulated as the planning of collision-free paths between specified points without satisfying the optimization criterion, given a description of the robot’s target starting point, target ending point, and surrounding environment [5]. Path planning is closely linked to whether a mobile robot can achieve autonomy, and the planned path also determines whether the mobile robot is energy efficient and reduces losses. Multi-robot path planning involves the path planning of several robots and the collaboration between them, which is more complex [6,7,8,9]. However, single-robot path planning is the basis of multi-robot path planning. At the same time, the information about the robot’s working environment is often complex and changeable. According to the degree of mastery of environmental information, path planning can also be divided into global path planning with all environmental information known and local path planning with environmental information completely unknown or partially known [10]. From the planning results, both types of algorithms need to find the path from the starting point to the target. However, global path planning involves finding a feasible path on an established global map, with the purpose of evaluating the optimal criteria, while local path planning refers to the robot using carried sensors to construct a local map and plan a local feasible path. The primary goal is to avoid dynamic obstacles and has higher real-time flexibility.

The global path planning of a single robot is a more basic problem and is also the most universal. Since 1980, scholars have been working on global path planning. In the beginning, it was only necessary to study a path that would lead to the target endpoint. However, with further research, the requirement goal has changed from just needing to find a path that leads to the target endpoint to a comprehensive design that combines the consideration of optimization criteria, such as how to make the path shorter or cover a larger area, among other problems [6].

To solve the path-planning problem, scholars proposed various mobile robot path-planning algorithms. Traditional path-planning algorithms mainly include the artificial potential field method (APF) [11], element decomposition method [12], and graph search algorithm [13]. However, when the obstacles are complex, there are many drawbacks, such as the algorithm requires a large number of calculations; can easily fall into a local optimum; and produces a path that is not smooth and can easily have sharp points, which does not conform to the actual situation and increases the workload of mobile robots [14,15].

Among the metaheuristic algorithms, PSO has a relatively simple structure and is easy to implement, and thus, it is widely used [16]. The improvement of PSO generally focuses on the adjustment of the population structure and the optimization of the velocity and position update formulas [17]. Burman R. et al. [18] proposed a democratic-inspired particle swarm optimizer that uses the concept of companion groups to improve the convergence speed. Zhao et al. [19] introduced a nonlinear recursive function to adjust the inertia weight to avoid falling into local optimal solutions and increase the diversity of the particles. Yu et al. [20] proposed a new type of hybrid particle swarm optimization (PSO) algorithm, namely, SDPSO. Pozna et al. [21] proposed a hybrid metaheuristic optimization algorithm that combines particle filter (PF) and particle swarm optimization (PSO) algorithms. Mohammed Hussein et al. [22] proposed an improved particle swarm optimization (PSO) algorithm named MPSO. Liu et al. [23] proposed a hybrid path-planning algorithm based on optimized reinforcement learning (RL) and improved particle swarm optimization (PSO). However, the improved algorithms still have some limitations, such as low convergence accuracy and early maturity.

Due to the increasingly complex working environment of mobile robots, many existing path-planning methods have certain problems, such as long paths and low efficiency [24,25]. Previous related research work mostly used traditional path optimization methods. Although there was also related path-planning research based on the particle swarm algorithm, the classical particle swarm algorithm can easily fall into a local optimal solution and the path-planning effect is not good. In order to fully plan the collision-free, shortest, and optimal path of mobile robots, this study first adjusted the inertia weight parameter of particle swarm algorithm and sorted the position vectors of particles to improve the global optimization ability of the algorithm. Second, artificial potential field method is introduced to combine traditional methods with heuristic algorithms and meet actual obstacle avoidance needs to generate more reasonable mobile robot planning paths. The main contributions of this study were as follows:First, a particle swarm algorithm based on sorting optimization was proposed. By dynamically adjusting the inertia weight and sorting the position vectors of particles, the global and local optimization ability of the algorithm was balanced and the search performance of the algorithm was enhanced.Second, the artificial potential field method was introduced to combine this method with the improved particle swarm algorithm, improve the convergence speed and accuracy of the algorithm, and make the optimal path found more in line with actual needs.Third, this study conducted a comparative analysis of various path algorithms, demonstrated the effectiveness of the proposed algorithm, and experimentally analyzed the influence of the number of algorithm populations on the optimization effect, obtaining a suitable number of population settings.

The other parts of this paper are written as follows. The second part describes the construction of the path-planning model for mobile robots. The third part introduces the particle swarm algorithm, improved sequential particle swarm algorithm, and artificial potential-field-based particle swarm algorithm for path planning. The fourth part discusses the conducted experiments and analysis. Finally, the conclusion of this paper is presented.

## 2. Environmental Modeling and Problem Formulation

Mobile robot path planning is divided into two stages. The first stage involves building an environment model of the mobile robot motion space based on environmental information. The second stage involves using a path search algorithm based on the built environment model to find a path that can avoid obstacles and ensure that the robot moves from the target starting point to the target endpoint.

The common methods of environment modeling are the raster method, visual map method, and free space method. In comparison, the visual map method can obtain the shortest path. However, when the target starting point and target endpoint change, it is necessary to re-establish the viewable view. The algorithm is inflexible. At the same time, if there are more obstacles, the required amount of computation will be super large. The search efficiency is low and the searched path is not guaranteed to be optimal [26]. Therefore, the amount of research on path planning using this method is lower [27]. The free space method is a popular approach for modeling simple environments due to its simplicity and ease of implementation. It also has the advantage of not requiring the reconstruction of free space when the target starting point and target endpoint change. However, in complex environments, the modeling of the free space method can be challenging. As a result, academic research conducted on using the free space method for environmental modeling has been lower compared with the previous two methods. The raster method was selected for this study due to its low computational effort and its ability to provide a more intuitive representation of the barrier environment. This method overcomes the disadvantages of the previous two methods.

The raster method is a technique used to describe the mobile robot motion environment using a raster map. Each raster in the raster map can be assigned either free or obstacle attributes. The raster method typically partitions the mobile robot’s surroundings into grid cells with binary information. Obstacle areas are represented by 1 and free areas are represented by 0 [28]. The raster map model has the advantages of being intuitive, simple, easy to analyze, and easier to implement, and many researchers have therefore used the raster method more often to describe the environmental space in which the robot moves.

There are two manifestations of the raster method: the Cartesian coordinate system and the ordinal method. These two forms are essentially the same [29]. In the Cartesian coordinate system, each cell corresponds to $\left(x,y\right)$ one by one [30]. In the ordinate marking method, a planar right-angle coordinate system is established, and the unit length on the coordinate axes is the size of the grid. The grid position is represented in right-angle coordinates [31,32]. The serial number marking method starts from the lower left raster of the spatial model, and each raster corresponds to a serial number, which is added in the order from top to bottom, and then from left to right.

The relationship between the raster number and the coordinates is given as follows (the environment space constructed in this example):(1)$$m=\left(x-1\right)\times N+y$$
(2)$$y=\mathrm{mod}\left(m,N\right)$$
(3)$$x=\mathrm{int}\left(m,N\right)+1$$
where *m* is the ordinal number of the current raster, *x* and *y* are the locations of the robot, and *N* is the size in one dimension of the environment space N×N.

Since any map can form a reasonable raster map as long as it is divided finely enough, this study explored the optimal path based on the raster map. The optimal path expression between the initial point *p_1_* and the endpoint *p_n_* is shown below:(4)$$\min f(P)=\sum_{i=2}^{\left|P\right|}d(p_{i},p_{i-1})P\subset W,P\cap O=\emptyset$$
where *P* is the set of path points to be found, *W* is the set of all raster points, and *O* is the set of points that form the obstacle. $d(p_{i},p_{i-1})$ is the Euclidean distance between two points. There is no intersection between the set of path points to be found and the set of obstacle points, i.e., it is guaranteed that the path cannot pass through the obstacle. This study found the optimal set of path points *P_best_* using the improved particle swarm algorithm.

## 3. Methodology

### 3.1. Particle Swarm Optimization

Particle swarm optimization (PSO) is a swarm intelligence algorithm that was proposed by Kennedy and Eberhart and was inspired by the foraging behavior of bird flocks [33]. A flock of birds is one of the biological populations in nature, and any behavioral activity is usually the result of a combination of individual behavior and group behavior.

The PSO algorithm starts by having a large number of candidate solutions. These particles are continuously moved in the search space to find the optimal solution (called the global best solution) [34]. In this process, each particle determines its movement in the search space. It does so by combining its current position with the historical best position. Additionally, it incorporates some random perturbations. When improved positions are found, these positions guide the movement of the swarm. The process is repeated many times in an attempt to find a satisfactory solution, which, although not guaranteed, is expected to find a global or near-global optimal answer at the end of all algorithm cycles.

The algorithm can be summarized in four main steps, which are repeated until the stopping condition is satisfied as follows:(1)Assign initial random positions and velocities to all particles in the search space.(2)Evaluate the fitness of each particle.(3)Update the individual and global best positions.(4)Update the velocity and position of each particle.

The PSO algorithm has been successfully applied in many fields, such as path-planning and function-optimization problems. It was shown that the PSO algorithm provides better results in a faster and easier way compared with other methods [35].

The most important part of this process is the particle velocity update equation, which is given below:(5)$$v_{id}^{k+1}=\omega v_{id}^{k}+c_{1}\zeta \left(p_{gd}^{k}-x_{id}^{k}\right)+c_{2}\eta \left(p_{id}^{k}-x_{id}^{k}\right)$$

The equation for updating the position of the particle is as follows:(6)$$x_{id}^{k+1}=x_{id}^{k}+\gamma v_{id}^{k+1}$$
where $\left(i=1,2\cdots ,m;d=1,2\cdots ,Q\right)$, $v_{id}^{k}$ is the *d*th dimensional speed of the *i*th particle, $x_{id}^{k}$ is the *d*th dimensional *k*th generation position of the *i*th particle, $p_{id}^{k}$ is the *d*th dimensional *k*th generation individual optimal position of the *i*th particle, $p_{gd}^{k}$ is the *d*th dimensional *k*th generation global optimal position of the swarm, $c_{1}$ is the inertia weight of a particle tracking its own historical individual optimal value, and $c_{2}$ is the inertia weight of a particle tracking the optimal value of the whole population. Appropriate adjustment of the learning factors $c_{1}$ and $c_{2}$ can expand the search space of the particle swarm and avoid falling into local optimal solutions. ζ and η are random constants that are uniformly distributed in the interval [0, 1] and are used to maintain the diversity of the population. γ is a constant factor added in front of the velocity for the position update, which is called the constraint factor and generally takes the value of 1 [36].

From Equation (5), we can see that the speed update of particles in the PSO algorithm includes three parts: (i) The speed part of particles at the last iteration, where this part shows that the speed of the particle update is influenced by the current speed, which can have the ability to balance the global search ability and local exploitation ability. (ii) The individual particle learning part, where the individual optimal solution represents the result of self-learning of individual particles; this part can prompt the individual. The individual optimal solution represents the result of self-learning of individual particles, and this part can motivate individual particles to perform a better global search and prevent individual particles from falling into local optimal solutions. (iii) The part where the individual particle learns about the population, which reflects the information sharing between individual particles in the whole population. Under the joint action of these three parts, the particles will learn from the individual and population behaviors, and constantly update their speed and position so that the particles tend toward a search direction. Thus, it is possible for the particle population to search for the global optimal position solution.

Assuming a population of n massless and volume-free particles moving in a Q-dimensional space, the particles are constantly socially learning and self-learning to continuously adjust their search direction and search speed, and then plan an optimal path.

Let each particle be $x_{i}=\left(x_{i1},x_{i2},\cdots ,x_{iQ}\right)$, where i=1,2,⋯,n. Each particle $x_{i}$ has an adaptation value that is determined by the objective function, the direction and distance the particle moves will be determined by a velocity by continuously iterating, and then the particle searches the solution space to find the current optimal particle based on the interaction between individuals and shared information. Then, each point $x_{i}$ in the search space can be imagined as a potential solution to the path-planning problem.

### 3.2. The Proposed Algorithm

#### 3.2.1. Particle Swarm Algorithm Based on Ranking

The PSO proposed at the beginning is no longer suitable for the current increasingly complex and high-dimensional obstacle environment. In order to give PSO better global planning and local exploitation capabilities, a sorting optimization strategy was proposed. The parameter of inertia weight is very important in PSO. When the value of the inertia weight is large, PSO has better global search ability and faster convergence speed, but the solution is often not optimal; when the value of the inertia weight is small, PSO has better local search ability and can obtain better solutions, but the convergence speed becomes very slow or even stagnant [37]. In order for PSO to produce better solutions, faster convergence speed, stronger global search ability, and stronger local exploitation ability, a linear decreasing adjustment is used for PSO, which can give the algorithm stronger global search ability in the early search and stronger local exploitation ability in the late search, which can expand the search space and make the particle local search for better results.

The formula for adjusting the inertia weights is as follows:(7)$$\omega_{iner}=InertiaMax-i\times \left(InertiaMax-InertiaMin\right)÷maxgen$$
where *i* is the current number of iterations, *i* will increase from 1 to the maximum number of iterations one by one, *InertiaMax* is the maximum inertia weight, *InertiaMin* is the minimum inertia weight, and *maxgen* is the maximum number of iterations. Based on experiments and experience, in order to further increase the global search capability of the algorithm, *InertiaMax* was set to 0.93. In order not to deviate too far from 0.9 (a generally taken value), *InertiaMin* was set to 0.8. It can be seen that $\omega_{iner}$ gradually declined from 0.93 to 0.8 as the number of iterations increased, thus achieving a dynamic balance between global and local searching.

If the value of the inertia weight is zero, the speed of particles in the next iteration only depends on the optimal position solution searched by the current individual particle. If the value of the inertia weight is not zero, the particles will keep exploring new regions; therefore, by adjusting the inertia weight so that it decreases linearly, this can balance the global and local search ability of the algorithm, and thus, the algorithm will obtain a better solution.

Based on (5), the velocity update equation with the introduction of inertia weights becomes (8):(8)$$v_{id}^{k+1}=\omega_{iner}v_{id}^{k}+c_{1}\zeta \left(p_{gd}^{k}-x_{id}^{k}\right)+c_{2}\eta \left(p_{id}^{k}-x_{id}^{k}\right)$$

In addition to the introduction of dynamically changing inertia weights $\omega_{iner}$, the position vectors in the PSO calculation process are ranked, which, in turn, leads to a PSO based on ranking (rPSO). The specific steps are as follows.

Step 1: Read the map data, where there are N×N nodes and the particles-per-node range is [0, 1].

Step 2: Set the parameters of the particle swarm algorithm. Based on literature research and experience, the population size was 50, the maximum number of iterations was 150, $c_{1}$ was 1.8, and $c_{2}$ was 1.7.

Step 3: Initialize the particle population.

Step 4: Update the particle velocity using Equation (8).

Step 5: Update the particle positions using Equation (6).

Step 6: Decode the particles to obtain the particle position vector.

Step 7: Sort the position vector of the particles. The PSO is used to optimize the random sequence of real numbers *p*, which, in turn, yields the effective encoding described above.

Step 8: Generate the path and calculate the total length of the path. After sorting the position vector *p*, sort *J* is obtained. The path between the starting point and the endpoint is obtained by sort *J*, and the path length is calculated.

Step 9: Calculate the fitness. If the iteration is completed, the result is output, and if the iteration is not completed, the process is returned to step 4.

The flow of the rPSO is shown in Figure 1.

#### 3.2.2. Particle Swarm Path-Planning Algorithm Based on Artificial Potential Fields

The artificial potential field method (APF) is a virtual potential field function method based on the electric field principle in physics, where “opposites attract, same charge repels”, which virtualizes the path planning of mobile robots in the environmental space as the motion in the force field [11]. The basic idea is to create a virtual force field in the working environment of the mobile robot, where the target point generates an attractive potential field and the obstacle generates a repulsive potential field. The mobile robot moves toward the target point under the combined effect of the attractive and repulsive fields. The artificial potential field *U*(*X*) and robot force *F*(*X*) equations are (9) and (10), respectively:(9)$$U(X)=U_{att}(X)+U_{req}(X)$$
where *U_att_*(*X*) is the attractive potential field generated by the target point, *U_rep_*(*X*) is the repulsive potential field generated by the obstacle, and *X* is the current position of the mobile robot;
(10)$$F(X)=F_{att}(X)+F_{req}(X)$$
where *F_att_*(*X*) is the attractive force of the target point and *F_rep_*(*X*) is the repulsive force of the obstacle on the mobile robot.

The potential field, in turn, generates a force and the robot moves in the direction of the combined force. APF is widely used in path planning because of its simple mathematical principle, low computational effort, high real-time performance, low hardware requirements, fast response time, and the ability to form closed-loop control for path generation and motion control with the working environment. In this study, the potential field of robot movement on the map was considered to be established first, and then rPSO was used for path planning. Therefore, a particle swarm path-planning algorithm for mobile robots based on an artificial potential field (apfrPSO) was proposed.

The apfrPSO algorithm obtains the position vector of each particle by encoding the real numbers, which correspond to the decision variables of the function, and then calculates the objective function value using the corresponding decoding method. It then continuously updates the velocity and position of the particles using rPSO to complete finding the optimal solution. The combination of APF and PSO uses the artificial potential field method to set a potential field value for each grid and uses the particle swarm algorithm to plan a path in which one potential field value tends to another smaller potential field value along the decreasing direction of the potential field in the synthetic potential field to plan a collision-free path from the starting point of the target to reach and stop at the end of the target. The specific process is as follows.

Step 1: Read the map data, where there are N×N grid points. The environment space where the mobile robot moves is divided into grids and each grid is set with the corresponding potential field value.

Step 2: Set the particle swarm algorithm parameters.

Step 3: Initialize the particle swarm.

Step 4: Update the particle velocity using Equation (8).

Step 5: Update the particle positions using Equation (6).

Step 6: Particle decoding, i.e., calculate the objective function to obtain the path length of each particle.

Step 7: Output the potential field, which is encoded as the potential field of each grid point with the position of the particle swarm to obtain the path.

Step 8: Generate the path and calculate the total length of the path if the potential field reaches the global minimum, that is, it reaches the target endpoint to get the path. If the iteration is completed, the result is output, and if the iteration is not completed, return to step 4.

The flow of apfrPSO is shown in Figure 2.

## 4. Numerical Experiments

In this experiment, the map size constructed using the raster method is 20 × 20, i.e., N = 20 (but 40 is used in classic terrain validation), and PSO mainly uses the suggestions given in the literature [37]. In this part, the research comparison and analysis were mainly conducted for rPSO and apfrPSO. In addition, a comparative analysis of other advanced algorithms was carried out.

### 4.1. Classic Terrain Validation

In this study, we used the raster method to model the environment of the path-planning space of the mobile robot and artificially set four types of obstacles (as shown in Figure 3 below). We set the location of the target starting point and the target ending point and observed whether the algorithm combining APF and PSO could meet the requirements and whether the path taken could avoid the set obstacles.

From Figure 3, it can be seen that apfrPSO was able to start from the target starting point and finally reach and stop at the target endpoint under these artificial randomly set obstacle environments and perfectly avoid all obstacles during the journey, that is, it was able to meet the basic requirements of path planning in the set environment space.

### 4.2. Algorithm Comparison Analysis

The program randomly generates an obstacle environment and artificially and randomly sets the target starting point and target ending point. The parameters are set uniformly under the condition of comparing the path length, the average time of the algorithm run (the algorithm was run 30 times and averaged) and the number of iterations when the algorithm reached the target optimal solution for both algorithms. The stochastic environment shown in Figure 4 is an example of a random environment.

First, the convergence performances of the two algorithms were compared and analyzed for the stochastic environment. As can be seen in Figure 5, the apfrPSO algorithm not only converged in a faster time but also obtained shorter paths in the path-planning problem compared with the rPSO algorithm. This meant that the apfrPSO algorithm had better convergence and global optimality search compared with the rPSO algorithm.

Then, a comprehensive comparison of the algorithms for multiple other stochastic environments was performed. The data in Table 1 and Table 2 show that in these four random obstacle environments, apfrPSO produced shorter paths and had shorter average running times than rPSO, all other conditions being equal.

From the data in Table 3, it can be seen that in these four stochastic obstacle environments, there was no fixed number of iterations corresponding to the optimal solution of the objective for both algorithms.

In addition to investigating the comparison between apfrPSO and rPSO for mobile robot path planning, the apfrPSO algorithm was also compared with other state-of-the-art algorithms. The experiments took place in the same four stochastic environments mentioned above. The other algorithms compared were DAFSA [26], IDAFSA [26], and IPSO-IDE [37]. DAFSA is a Dijkstra-based artificial fish swarm algorithm. IDAFSA is an improved Dijkstra-based artificial fish swarm algorithm, and IPSO-IDE is an improved particle swarm algorithm based on optimized differential evolution. Thus, experiments were conducted to verify the apfrPSO algorithm’s performance. A preliminary experimental study aimed at ensuring the fairness of algorithm comparison was therefore conducted, specifically to ensure that all algorithms had the same number of iterations. One can comprehensively value the results through iteration numbers. Thus, mainly each algorithm’s population size was set. For the other parameters of these algorithms, the reader is referred to the related references [26,37].

The planning results of the different algorithms for the paths are shown in Table 4. The table mainly compares the average path length and the average number of iterations (taken as integers). From the results, it can be seen that IPSO-IDE obtained slightly worse path lengths than apfrPSO for environments 1 and 3, and better path lengths than apfrPSO for environments 3 and 4, but IPSO-IDE had a higher overall number of iterations compared with apfrPSO. The overall performance of apfrPSO and IPSO-IDE was better than IDFSA and DAFSA. A comprehensive comparison showed that the apfrPSO algorithm had a strong convergence capability and was able to obtain very competitive path lengths.

### 4.3. Population Change study

In order to further explore the path optimization effect of the algorithm, the exploration performance of the algorithm with different population sizes was investigated. With the environment setup shown in Figure 4, the changes in path length and running times of the two algorithms were observed while only changing the population, with all other parameters unchanged, and the specific data are shown in Appendix A Table A1 (length 1 corresponds to the path length of rPSO, length 2 corresponds to the path length of apfrPSO, time 1 corresponds to the average running time of 30 runs of rPSO, and time 2 corresponds to the average running time of 30 runs of apfrPSO). The plot of the population size versus path length for both algorithms (Figure 6) and the population size versus average running time for both algorithms (Figure 7) were drawn from the data in Table A1.

From Figure 6, it can be seen that the path length curve of apfrPSO was more stable and the path length of apfrPSO was generally shorter than that of rPSO under the same circumstances. From Figure 7, it can be seen that the larger the population size, the longer the average running time of both algorithms, and the average running time of apfrPSO was shorter than the average running time of rPSO.

The relationships between the path length, average running time, and population size of apfrPSO in Figure 6 and Figure 7 show that the optimization capability was saturated after the population size reached a certain level and the path length cannot be further optimized, and thus, if the population size is increased at this time, it will make the algorithm run longer. Therefore, when designing a specific algorithm, the population size should be chosen reasonably so that a better result can be achieved in a shorter time.

## 5. Conclusions

Mobile robots are becoming more and more widely used in all aspects of our production life and path planning is another important part of the study of mobile robots. To address the problem of path planning for mobile robots, an algorithm that combines APF and PSO was proposed in this study to perform path planning for mobile robots on the premise of raster maps, and the process of the algorithm was introduced. In the simulation experiments, it was first shown that a collision-free path could be effectively planned from the starting point of the target to finally reach the end target. Then, the paths planned using rPSO and apfrPSO were compared through the experiments of various obstacle paths, and the results verified that the path planning of the mobile robot using apfrPSO had a shorter path and shorter running time than when using rPSO. In addition, the apfrPSO algorithm was compared with other state-of-the-art algorithms. The results show that the algorithm proposed was very competitive.

In the future, we aim to explore more intricate scenarios to validate the performance of the proposed algorithm and conduct real-world robot path-planning experiments. Solutions to problems such as local optimization and dead zones are also in our research plan.

## Figures and Tables

**Figure 1 sensors-23-06082-f001:**
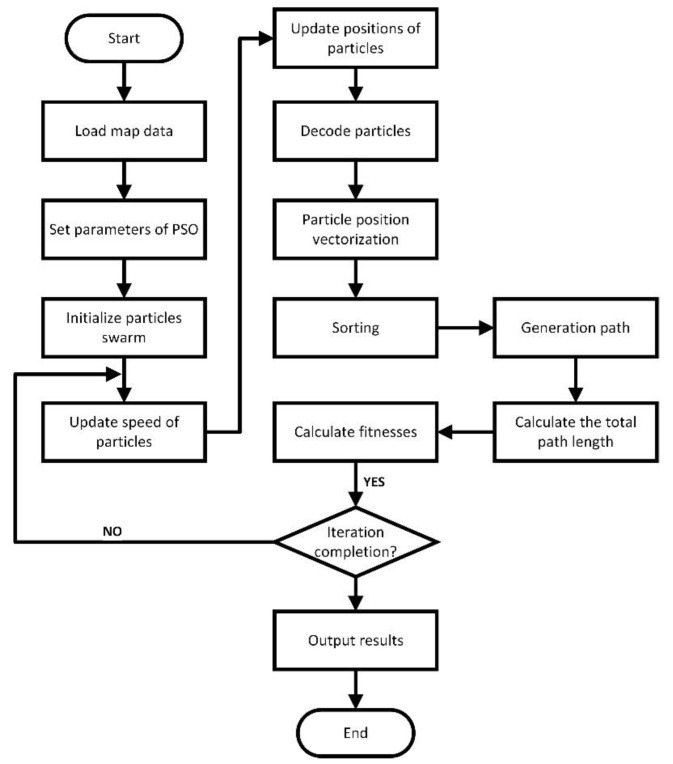
The flow diagram of rPSO.

**Figure 2 sensors-23-06082-f002:**
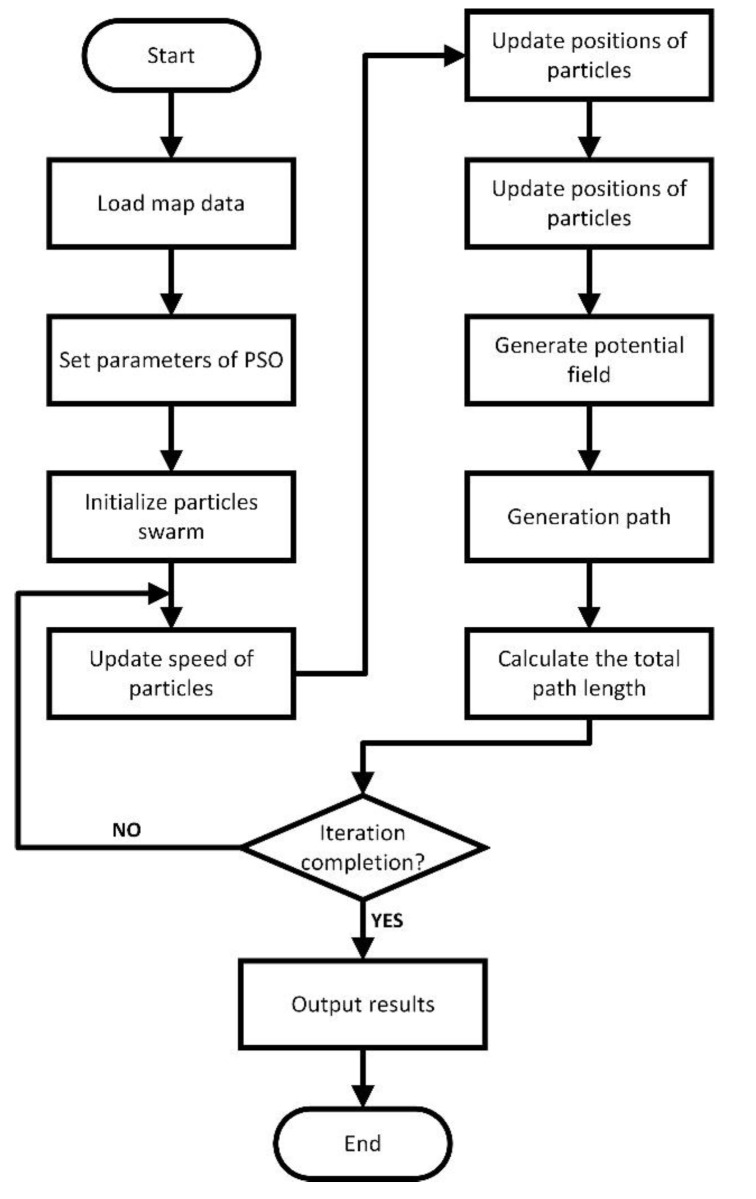
The flow diagram of apfrPSO.

**Figure 3 sensors-23-06082-f003:**
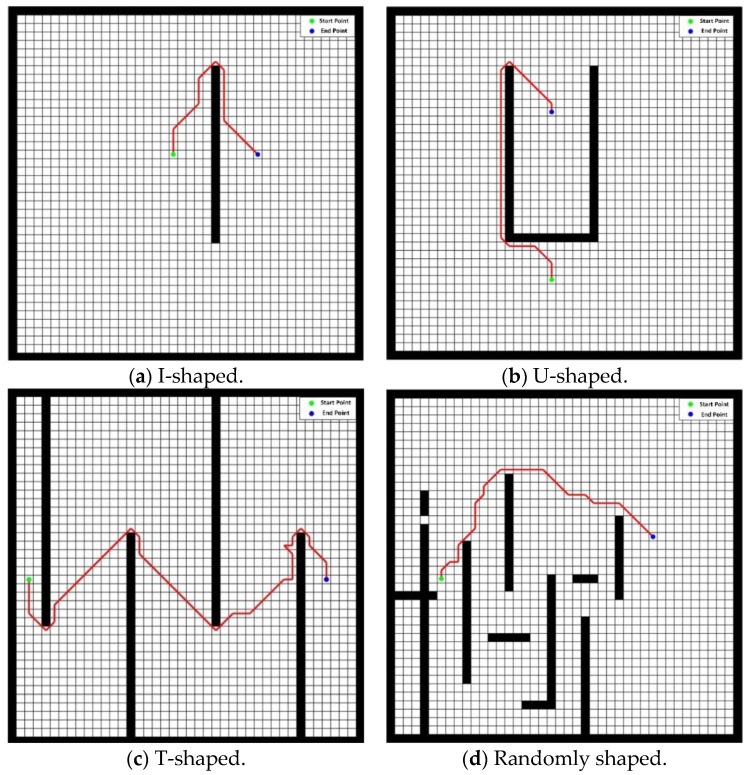
Simulation results in multiple obstacle environment types.

**Figure 4 sensors-23-06082-f004:**
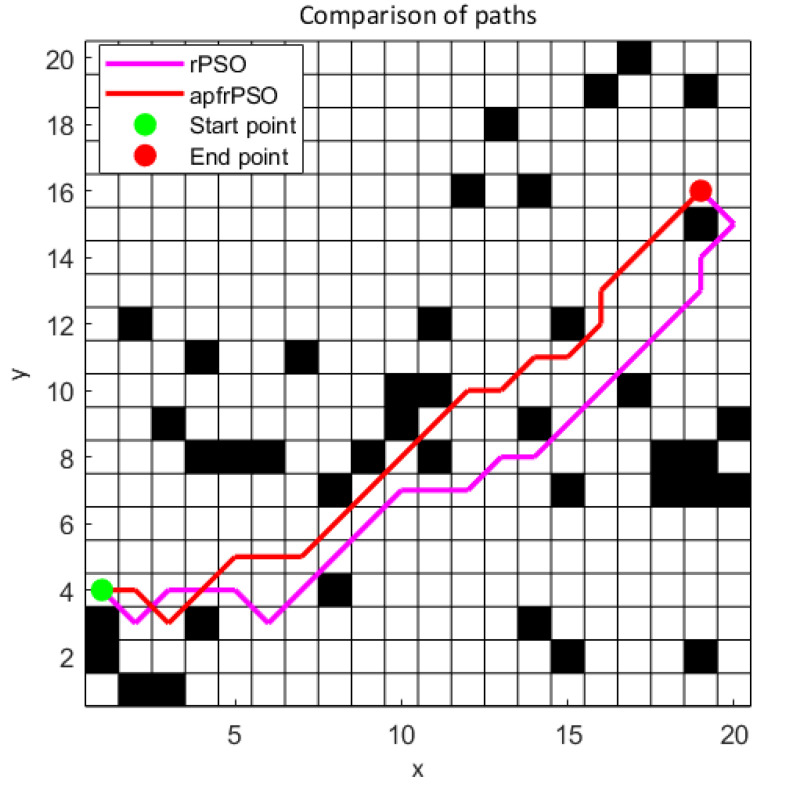
Simulation result graph of a random obstacle environment.

**Figure 5 sensors-23-06082-f005:**
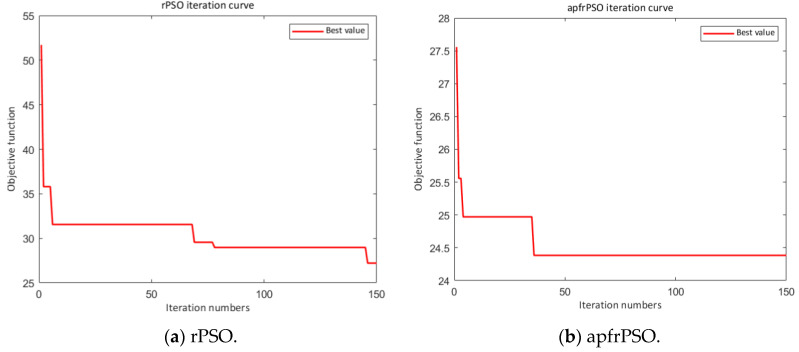
Comparison of iteration curves of the path optimization algorithms.

**Figure 6 sensors-23-06082-f006:**
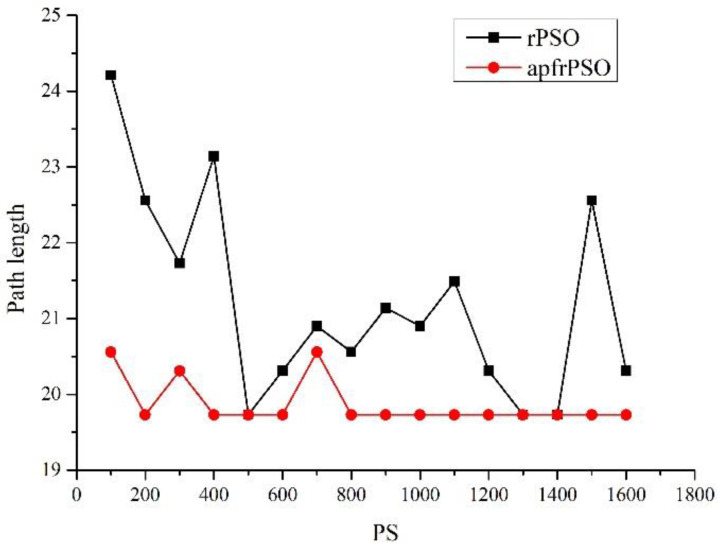
Population size versus path length found by the algorithms.

**Figure 7 sensors-23-06082-f007:**
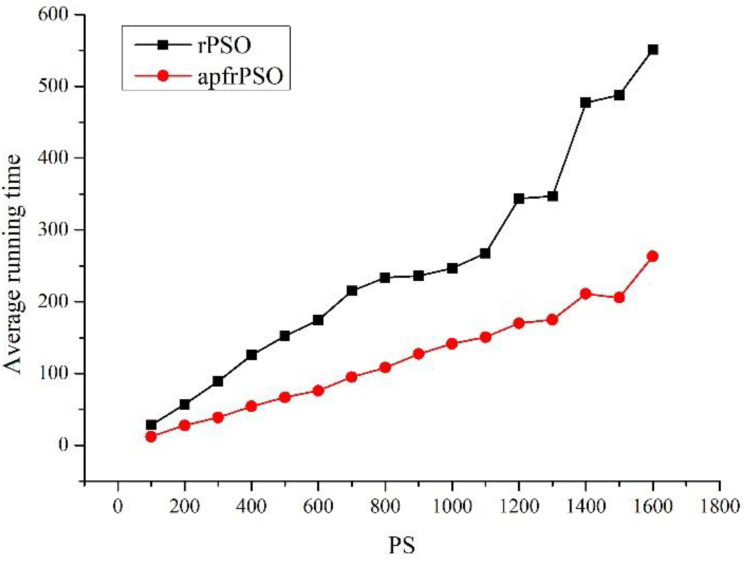
Population size versus the average running time of the algorithms.

**Table 1 sensors-23-06082-t001:** Comparison of the path lengths of the two algorithms for different obstacle environments.

Item	Environment 1	Environment 2	Environment 3	Environment 4
rPSO	27.21	24.97	22.14	21.14
apfrPSO	24.38	21.56	19.31	19.73

**Table 2 sensors-23-06082-t002:** Comparison of the average running times of the two algorithms for different obstacle environments.

Item	Environment 1	Environment 2	Environment 3	Environment 4
rPSO(s)	20.67	14.39	15.14	17.12
apfrPSO(s)	10.44	6.89	7.49	7.38

**Table 3 sensors-23-06082-t003:** Comparison of the number of iterations corresponding to the objective optimal solution of the two algorithms for different obstacle environments.

Item	Environment 1	Environment 2	Environment 3	Environment 4
rPSO	146	24	71	44
apfrPSO	36	30	2	48

**Table 4 sensors-23-06082-t004:** Comparison of apfrPSO with the other algorithms.

Item	Environment 1		Environment 2		Environment 3		Environment 4	
	Length	Iters	Length	Iters	Length	Iters	Length	Iters
DAFSA	28.32	91	26.46	68	25.67	78	24.12	75
IDAFSA	26.43	76	23.75	46	22.48	46	22.16	64
IPSO-IDE	24.49	43	21.24	38	20.16	10	19.24	53
apfrPSO	24.38	36	21.56	30	19.31	2	19.73	48

## Data Availability

Not applicable.

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
