# Peer review of "Particle Swarm Algorithm Path-Planning Method for Mobile Robots Based on Artificial Potential Fields"

_sensors, 2023, doi:10.3390/s23136082_

Round 1

Reviewer 1 Report

The manuscript studies the particle swarm algorithm path planning method for mobile robots based on artificial potential fields. The investigated topic is interesting and the manuscript has certain contributions. The paper can be refined by considering the following comments:

 1. The literature review in Introduction is not complete. First, when introducing that mobile robotics has been developed rapidly and has been applied in many fields by virtue of its high efficiency, independence from environmental restrictions and strong anti-interference ability, more latest relevant works can be introduced to support the statement, such as ‘Group-based distributed auction algorithms for multi-robot task assignment (2023)’ and ‘Multi-robot task assignment for serving people quarantined in multiple hotels during COVID-19 pandemic (2023)’. Secondly, when concluding that a reasonable planned path can greatly improve efficiency and reduce energy consumption, some closely relevant studies on obstacle-avoiding path planning need to be analyzed, such as ‘Path planning for wheeled mobile robot in partially known uneven terrain’ and ‘Distributed multi-vehicle task assignment in a time-invariant drift field with obstacles’. Furthermore, apart from the described algorithms, the optimal control theory-based algorithm is also popular for path planning as shown in references ‘An integrated multi-population genetic algorithm for multi-vehicle task assignment in a drift field’ and ‘Clustering-based algorithms for multivehicle task assignment in a time-invariant drift field’. These references are closely relevant to the path planning studied in the manuscript.

 2. Please clarify the main contributions of the manuscript in Introduction.

 3. A clear problem description or problem formulation is needed for better illustrate the studied problem. For example, it is not clear the objective function for the path plaThe manuscript studies the particle swarm algorithm path planning method for mobile robots based on artificial potential fields. The investigated topic is interesting and the manuscript has certain contributions. The paper can be refined by considering the following comments:

 4. In Numerical experiments, it is suggested to compare the designed path planning algorithms with some popular algorithms such as A* algorithm and Dijkstra’s algorithm in ‘Distributed multi-vehicle task assignment in a time-invariant drift field with obstacles’.

1. Some sentences are a little bit long, which affects the understanding of the paper.  

2. Please edit the manuscript to improve its readiness. 

Reviewer 2 Report

The authors proposed an artificial potential field-based particle swarm algorithm (apfrPSO) for mobile robots. The global path was created by adjusting the inertia weight parameter and ranking the position vector of particles, and then the Artificial Potential Field method is introduced to find the optimal solution of the particles. The paper successfully shows that their algorithm called apfrPSO performs better than the aPSO algorithm in path length and the average running time. 

However, there are significant concerns that I felt were not addressed well.

1. For the results/testing scenarios, the scenarios are relatively simple. I recommend the authors allude to a few examples from standard path planning techniques like narrow passage or mazes to find more interesting and complex (dynamics) scenarios.

2. The authors should discuss how apfrPSO solve the problems like local optimality and dead zone.

3. The authors should list the experimental results comparing apfrPSO with A*, D*, and so on.

In addition, the paper is rife with typos, incorrect notations, sentence constructions, and grammatical errors.

Reviewer 3 Report

This study proposes an Artificial Potential Field-based particle swarm algorithm (apfrPSO) for path planning of mobile robots in environments with obstacles. The algorithm combines rPSO and Artificial Potential Field methods to generate collision-free paths with shorter length and running time compared to traditional particle swarm algorithms, as demonstrated through simulation experiments on rasterized maps. The aim of the study is to improve the autonomy and intelligence of mobile robots by enhancing their path planning accuracy and efficiency.

1- The problem in the study and the review of the literature require correction. The problem in the literature should be explained one by one by emphasizing the shortcomings of the case studies and should be written as main contributions or highlights at the end of the introduction.

2- The study is based on simulation. For this reason, both traditional and competitor studies should be compared as methods. Traditional methods, methods and findings of competitor/similar studies and the method proposed here should be compared as a table. Parameters such as performance, number of robots, number of iterations, amount of errors, etc. should be used for benchmarking purposes in this table. For example, you can use the following studies:

https://doi.org/10.3390/pr11010026

https://doi.org/10.1007/s13369-020-04784-0

https://doi.org/10.3390/s22145217

https://doi.org/10.1038/s41598-021-04506-y

3- In order to prove that the work is not fabricated, stolen, dubious or plagiarized, access to the code, model, data and simulation should be given for blind review purposes. It would be nice if the authors did this for the readers as well.

Reviewer 4 Report

The proposed paper aims to improve the PSO algorithm with Artificial Potential Fields to find the globally optimal path for mobile robots from one position to another.

The paper is generally easy to follow, although there are minor language issues (stated below). The proposed method is not groundbreaking, but there is some degree of novelty.

I suggest the authors include more details about the APF method and applications (like DOI: 10.1016/j.ifacol.2022.04.211) to improve clarity. 

Some other minor issues:

- The citations are badly formatted in the text. Please look at (any) MDPI publication to find the correct citation formatting.

- Lines 178-180: authors describe both ζ and γ, but neither are used in equations (4) and (5). I guess ζ is to be used also in equation (7)?

- Equation (6): please format "maxgen" as a single variable, without the "max" Word equation markup

- Line 244: the PSO parameters are set: how are such values chosen? Is there a scientific reason behind "0.93 inertiaMax and 0.8 inertiaMin", or are "good" values driven by experience? 

- In equation (1) "N" is used. Then, in the description of the algorithms, "n" is used. The relation between n and N is n=N2? Please clarify this aspect.

- Authors have shown how apfrPSO rapidly comes to convergence. However, as can be noted from Figures 3c, 3d and 4, the paths could be further improved simply by removing unnecessary turns. Is there an algorithm explanation? 

- Please move Table 4 to be included in a single page

English language should be polished. I will provide some examples, but I suggest the authors carefully check the document.

- Line 57: "the" is missing the capital letter since it is placed after a "."

- Line 131: "from left to there" should be "from left to right"?

- Line 187: a full point is missing (before "The individual")

- Line 190: if the first two of "the three parts" are listed with "i" and "ii", then "third" would be substituted with "iii"

- Line 339: "environment1"

Round 2

Reviewer 1 Report

Thanks for the efforts of the authors in refining the manuscript, which can be accepted now. 

Author Response

Thank you very much for all the valuable comments you have provided for this article. We appreciate all the efforts have made.

Reviewer 2 Report

The manuscript has been effectively revised.

Please double check the grammar issues in the manuscript, such as Fig.3 and Figure 5.

Author Response

Thank you very much for your suggestion and we have checked and revised the paper carefully, including the issues you have mentioned above. Thank you very much for your revision work

Reviewer 3 Report

Overall, the revisions are satisfactory. But it would be better if performance benchmarking could be improved and detailed. Also, I think evidentiary content is essential, but I leave that decision up to the editor.

Author Response

Thank you very much for your valuable suggestions. Due to time constraints, we have only conducted preliminary verification of the algorithm's effectiveness in this work. However, we will continue to improve it in the future. We have made some modifications in the Conclusion section. Additionally, we try to communicate with our laboratory to open source the code in the future. The revision in Conclusion is as follows. 

"In the future, we aim to explore more intricate scenarios to validate performance of the proposed algorithm and conduct real-world robot path planning experiments. Solutions to problems such as local optimization and dead zones are also in our research plan."